# Pro-Apoptotic Effects of Anandamide in Human Gastric Cancer Cells Are Mediated by AKT and ERK Signaling Pathways

**DOI:** 10.3390/ijms26052033

**Published:** 2025-02-26

**Authors:** Víctor M. García-Hernández, Ana Laura Torres-Román, Erika Ruiz-García, Abel Santamaría, Joaquín Manzo-Merino, Alejandro García-López, Ruth Angélica-Lezama, Juan A. Matus-Santos, Oscar Prospéro-García, Julián Navarro-Ríos, Alette Ortega-Gómez

**Affiliations:** 1Translational Medicine Laboratory, National Cancer Institute, S.S.A., Mexico City 14080, Mexico; soopvictor@hotmail.com (V.M.G.-H.); torresana@ciencias.unam.mx (A.L.T.-R.); betzabe@yahoo.com (E.R.-G.); 2Nanotecnology and Nanomedicine Laboratory, Metropolitan Autonomous University-Xochimilco, Mexico City 04960, Mexico; absada@yahoo.com; 3Chemistry Science Faculty, Benemérita Universidad Autónoma de Puebla, Puebla 72570, Mexico; jmanzomerino@gmail.com; 4Biochemistry Unit, National Institute of Nutrition and Medical Science Salvador Suvirán, S.S.A., Mexico City 14080, Mexico; qfb_alejandro_garcia@hotmail.com; 5Cytology Laboratory, National School of Biological Sciences, National Polytechnic Institute (IPN), Mexico City 11340, Mexico; ralezama@hotmail.com; 6International Oncologic Center (COI), Mexico City 04700, Mexico; jams7878@hotmail.com; 7Cannabinoids Laboratory, Department of Physiology, School of Medicine, National Autonomous University of Mexico (UNAM), Mexico City 04510, Mexico; opg@unam.mx; 8Basic Investigation Department, National Cancer Institute, S.S.A., Mexico City 14080, Mexico; jnavarro_jnr@hotmail.es

**Keywords:** gastric cancer, anandamide, anti-proliferative, cannabinoids, apoptosis

## Abstract

Gastric cancer is one of the most common forms of cancer worldwide. A growing number of studies have addressed the anti-proliferative effects of cannabinoids on several tumor cells. The molecular mechanisms underlying the anti-proliferative effects of the endogenous cannabinoid anandamide (AEA) on gastric tumor cell lines have yet to be characterized. Here, we investigated the anti-proliferative mechanisms elicited by AEA on the AGS human gastric cancer cell line employing an Oncoprint database, Western blotting, and immunofluorescence. We observed that AEA (5 µM) inhibited phosphorylated AKT’s expression level. This point is relevant because AKT is mutated in AGS cells, according to Oncoprint. In addition, AEA induced the up-regulation of phosphorylated ERK and, in turn, inhibited Bcl-2 expression and activated pro-apoptotic signals induced by pro-apoptotic Bax and Bak, which resulted in caspase-3 activation. The effect of anandamide on phosphorylated AKT was dependent on cannabinoid receptor 2 activation (CB2R) as revealed by the selective inverse agonist JTE-907, which reverted the anandamide-induced expression in the phosphorylated AKT/total AKT ratio. In contrast, changes in phosphorylated ERK evoked an increase in pro-apoptotic pathways that culminated in cell death by caspase-3 activation. These results indicate that the endogenous cannabinoid anandamide in gastric cancer cells increases caspase-3 activity via mitochondrial pro-apoptotic Bax/Bak proteins and decreases viability through CB2R via AKT down-regulation’s trophic mechanisms. These effects constitute a promising tool for the design of gastric cancer therapies.

## 1. Introduction

Gastric cancer occupies the fifth position in incidence and is the third most prevalent cause of mortality worldwide [1]. Although surgery and chemotherapy are the standard treatments for this disease, these therapies are poorly effective, since they have been associated with high toxicity and low specificity. Because of the ineffective response that gastric cancer patients show for standard treatments, it is imperative to explore novel antitumor treatments. The deregulation of the endo-cannabinoid system (ECS) has been associated with several diseases, including cancer [2]. The anti-proliferative effects of cannabinoids on tumor cells have been reported in several in vitro studies [3], suggesting that the stimulation of the ECS by cannabinoids is responsible for the regulation of signal transduction pathways implicated in survival and cell death in various types of cancer cells [4]. In addition, the ECS is involved in the control of normal gastric motility, secretion, and the mucosal integrity of the gut [5]. Cannabinoid receptors comprise the classical cannabinoid receptor 1 (CB1R) and 2 (CB2R), as well as some receptors like G-protein receptor 55 (GPR55), the peroxisome proliferator-activated receptor (PPAR), the vanilloid receptor (TRPV), the TRP ankyrin channel (TRPA), and the TRP melastatin channel (TRPM) [4]. Although CB1R and CB2R are canonical receptors, there is significant overlap between cannabinoids and agonists. All these receptors are important translators of cannabinoid actions in the gastrointestinal tract and critically determine the course of cancer progression [5]. It is known that CB1R and CB2R activation participates in the modulation of survival, stress, invasiveness, metastasis, and metabolism in different cancer models [3,5,6,7,8,9,10]. While some recent studies have shown that CB1R is expressed in the normal colonic epithelium and colorectal cancer cell lines the peripheral CB2R is mostly expressed in the immune system [5]. Actually, there are reports available in the literature of cannabinoid and anti-proliferative activity. Most of these studies are related to phyto-cannabinoids, associated with addictive side effects. Because of the new evidence about endogenous cannabinoid stimulation and anti-tumoral activity in various types of cancer, it is relevant to study the intracellular mechanisms that sustain this effect on gastric cancer. At this point, the anti-proliferative effects of endogenous CB1R and CB2R agonists in different tumor cell lines have been attributed to the increase in the sphingolipid ceramide and the subsequent activation of deadly cascades [4]. Other anti-proliferative effects of cannabinoids have been related to the activation of multiple cell death programs, depending on the tumor cell type. Among these mechanisms, the inhibition of AKT has been directly associated with the ability of some cannabinoids to control cell cycle checkpoints in a CB1R-dependent manner. The well-known endo-cannabinoid and CB1R agonist N-arachidonoyl-ethanolamine (anandamide) is synthesized from arachidonic acid [2] and plays important physiological and anti-proliferative roles in organisms. It is known that cannabinoids have pharmacological synergism effect with paclitaxel in gastric cancer cell lines [11]. AEA can reduce the volume of a cholangiocarcinoma tumor in an in vivo model by up-regulating Wnt5a expression via the tyrosine kinase-like orphan receptor (Ror2) and c-Jun-N-terminal kinase (JNK) pathway regulation [12]. Tumor growth inhibition induced by AEA also involves GPR55 receptor activation, as well as the down-regulation of several angiogenic factors. It is noteworthy that these effects were also observed using the AEA analogous Met-F-AEA or the fatty acid amide hydrolase (FAAH) inhibitor URB597 [12]. In addition, it is known that the selective CB1R antagonist AM251 revert the anti-proliferative effect of AEA [4]. Moreover, colorectal cancer (CRC) proliferation has been shown to be mediated by the increased expression of CB1R through the transcriptional regulation of the *CNR1* gene promoter [4]. In turn, the phyto-cannabinoid CBD is known to regulate the p-AKT pathway in gastric cell lines [13]. Consequently, the pro-apoptotic effects of CBD in gastric cancer have been associated with ROS production by mitochondria [14]. There is also evidence that THC induces apoptosis via CB1R through the inhibition of the phosphoinositide 3-kinases-AKT-PI3K signaling pathway in colon cancer [15]. Autophagy has also been demonstrated by CB2R agonists such as JWH-015, with consequent AKT/mTOR inhibition and AMPK activation. Thus, it is generally accepted that cannabinoid agonists act on tumor cells through several mechanisms, such as follows: (a) suppressing the production of inflammatory cytokines such as interferons (IFN-α, -β, and γ), tumor necrosis factor (TNF), and inteleukin-1β [16]; (b) acting upon canonical receptors such as CB1R and TRPV1 and regulating the tropomyosin receptor kinase (TRK) receptors implicated in cancer development [16]; (c) inhibiting cell proliferation and colony formation through p53 inhibition and cell cycle arrest (G0-G1) by the inhibition of p21, CDK2, and cyclin E proteins [17]; (d) inducing cell death by Bax/Bcl-2 and the inhibition of mitochondrial membrane potential (MMP); (e) generating reactive oxygen species (ROS) associated with G0/G1 cell arrest and apoptosis [18]; (f) suppressing the X-linked inhibitor of apoptosis protein (XIAP) and increasing the ubiquitination of XIAP [16]; (g) inducing cell cycle arrest in G0/G1 and ERK inhibition [17]; (h) inhibiting migration, invasiveness, and the epithelial–mesenchymal transition (EMT) evoked by AKT through the down-regulation of cyclooxygenase 2 (COX-2) [19].

Previously, we studied the effects of AEA and other synthetic cannabinoid agonists on an in vitro gastric cancer model. We observed that AEA induced anti-proliferative effects associated with morphological changes, cellular viability inhibition, DNA fragmentation, and cell death in AGS cells [20]. As a logical extension of our previous work, the aim of this study was to evaluate signal transduction pathways involved in the anti-proliferative and cell death mechanisms in gastric cancer tumor cells treated with AEA, thus strengthening our knowledge in regard to the mechanisms triggered by endo-cannabinoids as signal transduction effectors. For this purpose, we investigated the signaling pathways involved in AEA-induced anti-proliferative effects in AGS cells and established the mechanistic pathway through which AEA exerts its effects on AGS cells in vitro. Our results suggest that AEA modulates the ECS to induce antitumor effects and may be a potential candidate for the design of therapeutic strategies against gastric cancer.

## 2. Results

### 2.1. Activation of CB2R Induced Down-Regulation of p-AKT Expression in Time-Dependent Manner

In a previous report, we found that AEA (5 μM) reduced cell viability and increased cell death in AGS cells [20]. Here, we investigated whether these effects of AEA (5 μM) were linked to changes in p-AKT expression. AEA decreased the levels of the p-AKT (p-AKT/total AKT) ratio at different times (0, 2, 6, 12, and 24 h) (Figure 1A), reaching a significant effect at 6 h (*p* ≤ 0.05, different to the control), 12 h, and 24 h (*p* ≤ 0.01, different to the control). We also evaluated the involvement of CB1R and CB2R in the AEA decrease in p-AKT in AGS cells using classical antagonists for these receptors (Figure 1B). For this purpose, AGS cells were pre-treated for 1 h with AM281, AM251, or JTE-907 and then exposed to AEA (5 µM) for 24 h. As revealed by immunoblot analysis (Figure 1B), the CB1R inverse antagonists AM281 and AM251 failed to prevent the AEA-induced decrease in the p-AKT/total AKT ratio, whereas the CB2R antagonist JTE-907 restored this ratio in a significant manner (*p* ≤ 0.01, different to AEA) (Figure 1B).

### 2.2. AEA Induced Changes in Immune Localization of p-AKT in AGS Nucleus

Next, we explored p-AKT cytoplasm/nucleus immune localization. For this purpose, AGS cells were treated with AEA (5 µM) and analyzed, using triple-staining, phalloidin (left column), p-AKT (middle–left), and nuclear DAPI (middle–right) staining (Figure 1C). AGS cells treated with AEA (5 µM) for 24 h showed a diminished amount of nuclear p-AKT compared to the control. This effect was partially reverted by the CB1R antagonist AM281 (1 µM). The CB1R antagonist AM251 and JTE-907 had no effect on the AEA and p-AKT immune location in the AGS cell line.

### 2.3. Oncoprint Query “Endocannabinoids Genes Events” and Comparisons Between Esophageal/Stomach Cancer Cell Lines

The query confirmed that AKT is abnormal in gastric cancer cell lines; specifically, the AGS cell line showed an amplification in the AKT2 gene (Figure 2) whereas other esophageal/stomach cell lines express other significant gene variations as missense mutations or deep deletions. Specifically, Oncoprint analysis showed no variations in the endocannabinoid genes’ modulation system (CNR1 (or CB1R), CNR2 (or CB2R), and FAAH (AEA degradation enzyme)).

### 2.4. AEA Increased in p-ERK and Pro-Apoptotic Signals Through Caspase-3 in AGS Cells

Under certain conditions, the overstimulation of ERK1/2 signaling can cause tumor cell death (72). Next, we examined the effect of ERK over other pro-apoptotic proteins in AGS cells exposed to AEA (5 µM) for 12 and 24 h. AEA increased p-ERK expression in a time-dependent manner (Figure 3A). The levels of the p-ERK (p-ERK/total ERK) ratio had a significant effect at 2, 6, 12 (*p* ≤ 0.05, different to the control), and 24 h (*p* ≤ 0.01, different to the control) (Figure 3A). Simultaneously, Bcl-2 expression was decreased by AEA at 12 and 24 h of incubation (*p* ≤ 0.01, different to the control). In contrast, both the Bak and Bax relative expressions were increased at both times tested (*p* < 0.05, different to the control, respectively) (Figure 3B). Bcl-2 inhibits apoptosis through the preservation of mitochondrial membrane integrity as its hydrophobic carboxyl-terminal domain is linked to the outer membrane. Altogether, these results suggest that AEA overstimulates ERK and trigger Bax and Bak signaling in the outer mitochondrial membrane to increase the permeability of pro-apoptotic molecules derived from the mitochondrial inner compartment, thus activating intrinsic apoptotic cell cascades.

Finally, we evaluated cleaved caspase-3 as a final effector for apoptosis cell death in the AGS cell line; therefore, AEA induced protease activation in AGS cells initially through procaspase-3 fragmentation. The p17 fragment result of caspase-3 fragmentation significantly increased its expression in AGS cells exposed to AEA in a time-dependent manner (12 and 24 h), compared to the control (*p* ≤ 0.01) (Figure 4A). The protein kinase inhibitor staurosporine (STS; 1 µM) was used as a positive control for caspsase-3 activation and cell death, exhibiting increased levels of these endpoints at 2 h of incubation (*p* ≤ 0.01; different to the control) (Figure 4B).

## 3. Discussion

In the past two decades, studies on gastric cancer have addressed the propensity of the ECS to activate cell death programs as a potential target site for novel pharmaceutical approaches [11,17,20,21,22]. Plants and synthetic cannabinoids are known as anti-proliferative drugs in several cancer types, including gastric cancer; however, the characterization of mechanisms related to this effect is not well known and requires systematic investigations. Here, we demonstrated the intracellular mechanisms underlying the anti-proliferative effects induced by AEA in the AGS human gastric cancer cell line, which appear to have been related firstly to a down-regulation of p-AKT and secondly to an up-regulation of p-ERK at times as short as 6 h in both cases, accompanied by the triggering of pro-apoptotic signals and silencing of survival pathways; all effects could be mediated by CB1R and CB2R, without discarding other pathways. These effects could modify cell signaling involved in survival and proliferative mechanisms in gastric tumor cells, as has been reported for survival pathways associated with PI3K/AKT and cell cycle effects through MAPK signaling regulation [23]. In this regard, it is also known that PI3K/AKT signaling provides metabolic energy for the synthesis of macromolecules that sustain the survival and aberrant maintenance of tumor cells [24], whereas the MAPK pathway activates oncogenes, cell motility, and duplication by the regulation of mitosis [25]. Therefore, Oncoprint analysis confirmed the AKT gene’s alterations in the AGS cell line in comparison with other gastric cancer cell lines. Other modifications, such as missense mutations and deep deletions, are associated with oncogenic phenotype and proliferation [26,27]. Cannabinoids’ effects are specific in tumor cancer cells with or without AKT amplification. Other mutations different from AKT could act as the main effectors of malignancy, such as KRAS, p.Gly12Asp (c.1357G>A), PIK3CA, p.Glu453Lys (c.1357G>A), and TP53 (1370612). In our study, AKT amplification could be relevant in AEA-treated cells, but it must be more studied.

Mitosis signals’ stimulation suppresses cell survival and activates apoptotic cell death [28,29,30,31,32]. Several studies have suggested that cell death activation is cell cycle-dependent [33,34]. In this regard, mutagenic signals are known to activate metabotropic receptors associated with the G0/G1 steps of the cell cycle in several types of tumor cells lines. G0/G1 activity then induces the accumulation of p27/Kip-1 and p21/Cip-1, inhibiting CDKs-dependent kinases that induce cell tumor arrest [35,36]. Other studies have reported that CB1R activation by WIN-55,212,2 increases the activity of p-ERK1/2 by up-regulating the kinase-dependent cyclin inhibitors p27/Kip-1 and p21/Cip-1/WAF, thus inducing a decrease in D1, E, and CDK-2,4-6 cyclins, promoting cell cycle arrest in GO/G1 phases [17]. In support, clinical data show that AEA in combination with paclitaxel or 5-FU induces anti-proliferative effects through apoptosis in gastric cancer cell lines involving ceramide activation and probably changes in the cell cycle, observed at 48 h [11]. In line with this, it is known that PI3K/AKT is one of the main survival signaling involved in gastric cancer tumor progression. Several studies have reported that cannabinoids may induce the modulation of ERK1/2 and AKT activity in gliomas and prostate cancers [37]. The regulation of both AKT and ERK activities in tumor cells depends on the lasting and nature of the stimuli, as well as on the tumor cell type. Thus, the main signaling pathways involved in anti-proliferative effects activated by cannabinoids regulate migration, proliferation, survival, and cell death [24,25]. In a previous report, we demonstrated the anti-proliferative effects of AEA at different concentrations (0.5, 2.5 and 5 µM), evaluating morphological, biochemical, and nuclear changes associated with apoptosis in an AGS cell line [20]. As a logical extension, we expanded our knowledge of cannabinoids’ anti-proliferative effects, studying intracellular mechanisms associated with tumor survival/death by exploring MAPK signaling.

Several groups have demonstrated that prolonged ERK activation can mediate cell death [38,39,40]. In turn, the maximal activity of p-ERK is obtained during the first few minutes of stimulation, with a second early activity wave occurring later, where p-ERK is translocated into the nucleus, activating several genes [41,42,43,44]. In turn, the up-regulation of p-ERK increases the transcriptional activation of Ets2, C/EBP-α, and C/EBP-β, inducing cell alterations by p21/Cip-1 [45]. This evidence is consistent with our findings, where p-ERK up-regulation induced by AEA showed concentrations (IC_50_ = 5 μM) with time-dependent effects (from 2 to 24 h).

Our results demonstrate that CB1R and CB2R are both involved in the anti-proliferative effects of AEA in AGS cells. CB2R seems to regulate the effects of AEA, as demonstrated by the effect of JTE-907 on the AEA-induced changes in the p-AKT/total AKT levels. Taken together, we therefore hypothesize that the activation of both CB1R and CB2R, combined with other signaling mechanisms, is necessary to simultaneously activate the anti-tumor properties of AEA, converging signaling pathways to reduce tumor cell survival [46,47]. Therefore, we demonstrated an alternative anti-proliferative mechanism of AEA, combining the co-activation of CB1R and CB2R and compartmentalizing the regulation of AKT and ERK to evoke early pro-apoptotic cell signals in AGS cells. The regulation of AKT and ERK is known to be also related with the induction of stress signals in DNA stimulated by AEA and increased apoptotic cell death [20,48]. In the present study, apoptotic cell death occurred at 12 and 24 h after AEA treatment. It is noteworthy that during the same time period (12–24 h), the anti-apoptotic protein Bcl-2 decreased in AGS cells, whereas the pro-apoptotic Bax/Bad proteins increased, thus supporting the activation of intrinsic cell death programs. In line with these findings, we also found that AEA activated caspase-3 at 24 h, reinforcing the concept of programed cell death. The latter suggests that AEA can stimulate cell death, recruiting changes in mitochondrial activity. It is known that mitochondrial signaling participates in cell death induced by cannabinoids in gastric cell lines [14,18,49] since the accumulation of pro-apoptotic molecules increases the external mitochondrial membrane’s permeability [50]. While in cancer cell lines Bcl-2 interacts with Bax/Bak, inhibiting apoptosis [51], Bax/Bak accumulation in the external mitochondrial membrane induced by cytotoxic conditions is inhibited when Bad is attached to Bcl-2 or co-activators from the BH3 family’s domains (Bid, Bim, PUMA, etc.) [52]. Bax/Bak output from the inner mitochondria increases transition mitochondrial pores (MMP), which are necessary for releasing pro-apoptotic molecules for pro-caspase activation, thus activating apoptosis in tumor cells [53,54]. In turn, p-Akt phosphorylates Bad, retaining it in the cytoplasmic domain; next, SDK-1 kinase activates 14-3-3 protein which in turn releases Bad, increasing the external mitochondrial membrane’s permeability and releasing Bax/Bak [55,56]. Further, we noted that p-AKT was down-regulated in AGS cells in response to AEA, which might have kept Bad separated from Bcl-2, thus triggering apoptosis mediated by Bax/Bak. It is also known that p-ERK is directly involved in apoptosis by stress conditions releasing Bcl-2 and increasing Bax/Bak cytoplasmic levels [57]. Consistent with this concept, we found that Bax/Bak levels increased, while Bcl-2 levels decreased, thus favoring apoptosis.

At this point, we cannot discard the possibility that other mechanisms, independent of the events described herein, might occur in gastric tumor cells in response to AEA treatment. In this regard, ceramide signaling might also contribute to these anti-proliferative effects. Some studies have demonstrated that AEA promotes ceramide turnover to sphingomyelin on cell membrane [58] in a process closely related to TNF-α receptors [59]. p-AKT inhibition by AEA may involve phosphatase-2A (PP2A) activated by ceramides for cell death activation [60]. In addition, there is evidence that the anti-proliferative effects of AEA might also be related to its metabolic enzymes in tumor cells [6,61,62].

How these mechanisms contribute to the pathway described in this study, or vice versa, remains to be explored in further studies. In the interim, the evidence collected from this report and others clearly contributes to the concept that endogenous cannabinoids possess several anti-tumor properties deserving detailed characterization.

This study also supports the concept that, although AEA has been typically considered a CB1R agonist, it may also exert its effects via CB2R activation. It has been shown that AEA can be considered a partial CB2R agonist [63]. Several reports describe the important effects of AEA on CB2R; for instance, AEA decreases the lipopolysaccharide-induced release of nitric oxide in microglia via CB2R regulation [64]. In addition, through CB2R activation, AEA down-regulates the expression of the placental transporter by decreasing cAMP [3]. The effects of AEA on CB1R and CB2R depend on the extracellular concentration of endogenous cannabinoids, which in turn is physiologically regulated by its degradation enzymes and transport proteins [65]. Ionotropic receptors such as TRPM7 and TRPV1 are known to be involved in gastric cancer through the PLC/RAS/ERK pathway or through calcium loading via ERK/P38/JNK, activating cell proliferation or cell death [66,67,68,69]. Ultimately, the inhibition of the fatty acid amide hydrolase FAAH is a catabolic enzyme that supports antiproliferative effects on tumor cells via oxidative signals, such as NRF2 activation [70]. The physiological concentration of AEA in the gastric environment is around 1–5 nM [71]. It is known that endogenous cannabinoids (AEA) and receptors increase in immunitary tumor cells, known as the “immunitary endocannabinoid system” [71,72]. p-AKT acts through mTOR activation, increasing ribosomal proteins (S6K1 and 4EBP1) [73], and p-ERK 1/2 regulates transcription factors such as ETS, ELK, and Myc which direct cell death and metabolism [74]. The observed effects of cannabinoids on AGS cells were also studied on gastric xenografts, decreasing the tumor volume by 30% and decreasing invasion (MMP2/MMP9/MMP8) [75]. Our results demonstrated the activation of this pathway since AEA significantly increased caspase-3 cleavage and decreased Bcl-2 levels.

## 4. Materials and Methods

### 4.1. Materials

AEA (Cat. 1339), the CB1R antagonists/inverse agonists AM281 (Cat. No. 1115) and AM251 (Cat. No. 1117), the CB2R inverse agonist JTE9-907 (Cat. No. 2479), and the protein kinase C inhibitor staurosporine (Cat. No.1285) were obtained from TOCRIS Bioscience (Bristol, UK). Dimethylsulfoxide (DMSO; Cat. 0231) was acquired from AMRESCO (Solon, OH, USA). DMEM/F-12 1:1, Trypsin 0.25%, and fetal bovine serum (FBS) were purchased from GE Healthcare Life Sciences Hyclone Laboratories, Logan, UT, USA. All other reagents were of an analytical grade and obtained from known commercial sources.

### 4.2. Cell Culture

AGS human gastric adenocarcinoma cells (ATCC CCL-235, Grade IV) were purchased from the American Type Culture Collection (ATCC, Manassas, VA, USA). The cell line was initially cultured in T-75 bottles (Corning TM) containing DMEM/F-12 (1:1) medium supplemented with 10% FBS and 0.1 µg of penicillin and 0.1 µg/L of streptomycin; then, the cells were incubated at 37 °C and 5% CO_2_. Twenty-four hours later or until confluence, the cells were detached using 600 µL of Trypsin-EDTA 1X solution (Sigma Aldrich, St. Louis, MO, USA, 549430C) and re-seeded in 6-well plates treated with AEA (5 µM) (according to Ortega et al. [19]. Furthermore, the AGS cells were pretreated with CB1R or CB2R inverse agonists/antagonists; 1 μM of AM281 and AM251 were employed according to the literature [4,76,77,78]. On the other hand, 0.38 μM of JTE-907 was used as the literature indicates [79]. A total of 0.5 μM of staurosporine (STS; protein kinase C inhibitor) was used as a positive control for apoptosis, according to Ortega et al. [19,79].

### 4.3. Oncoprint Platform

Oncoprint is a database which provides information about genomic alterations for different tissue samples (patients, cell lines, and biopsies) or gene modifications (amplifications, deletions, insertions, and frameshifts). The Oncoprint data analysis of the gastric human cell line (AGS) included AKT1, AKT2, CNR1 (CB1R) and CNR2 (CB2R), and FAAH. The data analysis included gene alterations of other gastric cancer cell lines for comparative purposes to obtain more information about driver oncogenes in gastric cancer. Particularly, AKT amplification could act as an active guide in AGS tumor cells through a set of well-known driver genes, such as PIK3CA.

### 4.4. Western Blot Analysis

After all designed treatments with 5 µM of AEA at different times, the AGS cell line was washed with PBS solution and lysed with a RIPA lysis buffer system (Santa Cruz Biotechnology, Dallas, TX, USA, SC-24948) according to the manufacturer’s instructions. The final protein concentration was determined by Bradford’s method [80]. Equivalent amounts of protein from cell lines were separated by electrophoresis (SDS-PAGE) and transferred into a PVDF membrane. Then, the membranes were incubated with their corresponding primary antibodies (dilution 1:1000) overnight at 4 °C. Secondary conjugated antibodies with HRP used at 1:10,00 were directed against the following molecules: p-AKT (sc-514032; for three isoforms, AKT 1/2/3, serine 473, 474, and 472, respectively), total AKT-1 (sc-5298), p-ERK (sc-23759), total ERK (sc-271269), Bcl-2 (sc-7832), Bax (sc-20067), Bak (sc-517390), pro-caspase-3 (E8; sc-7272), caspasa-3 p-17 (sc-271028), and β-actin (H300; sc-10731; dilution 1:10,000) as a loading control (Santa Cruz Biotechnology, Dallas, TX, USA). Finally, the membranes were washed three times with TBST and bound antibodies were detected using chemiluminescence reagents (Pierce ECL, Western blotting substrate, Thermo Scientific, Waltham, MA, USA) and visualized using an immunoblot imaging system (Fusion FX5, Vilber Loumart, F-77202 Marne-la-Vallée Cedex 1, France). Bands were quantified by densitometry using Image-J software (1.49 version, National Institutes of Health, Bethesda, MA, USA).

### 4.5. Immunofluorescence

AGS cells at ${1.8\times 10}^{5}$ per well were seeded in 6-well plates on coverslips and maintained with DMEM/F12 medium at 37 °C in a humidified chamber with 5% CO_2_. After 24 h of culture, the cells were washed and treated with AEA (5 μM) alone or AEA treated previously for 1 h with CB1R or CB2R antagonists, and the plates were incubated for 24 h. Then, the cells were washed, fixed with PBS/PFA 4%, and permeabilized with PBS/Triton 0.1%, and the p-AKT (Santa Cruz Biotechnology, Dallas, TX, USA, sc; 514032) antibody was added and incubated overnight. The next day, the cells were washed and stained with Alexa 555-congujated anti-mouse (Molecular Probes, Eugene, OR, USA), and phalloidin 647 (Invitrogen, Eugene, OR, USA) for 2h. Furthermore, the nuclei were stained with 4,6-diamidino-2-phenylindole (DAPI, Thermo Scientific, Waltham, MA, USA) for 10 min at 37 °C in the dark. Finally, the cells were washed, mounted with mounting media (Vector laboratories, Burlingame, CA, USA), and observed using an EVOS FL microscope (Thermo Fisher Scientific Waltham, MA, USA) under a 40× objective.

### 4.6. Statistical Analysis

Results were expressed as the mean value ± S.E.M. or S.D. of a minimum of three independent experiments per group (each in triplicate). Data were statistically analyzed by one-way analysis of variance (ANOVA) for repeated measures, followed by Dunnett’s or Tukey’s tests. Values of *p* ≤ 0.05 were considered statistically significant. Analytical procedures were performed using the scientific statistical software GraphPad Prism 5 (GraphPad Prism Scientific, San Diego, CA, USA).

## Figures and Tables

**Figure 1 ijms-26-02033-f001:**
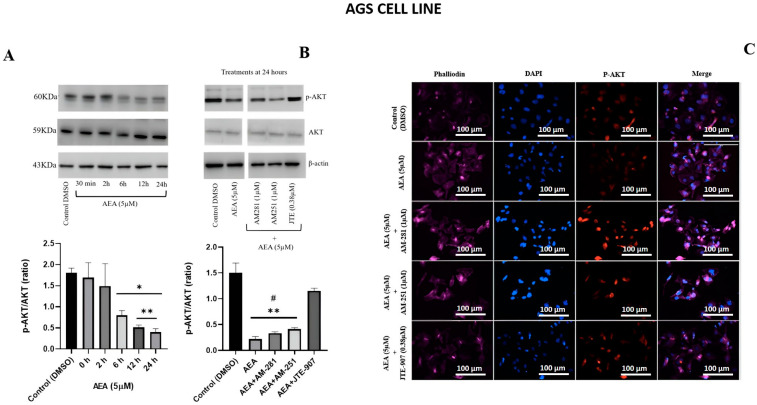
Changes in phosphorylated AKT (p-AKT) in AGS cell lines induced by AEA (5 µM). (**A**) The down-regulation of p-AKT induced by AEA (5 µM) at different moments of time (30 min, 2 h, 6 h, 12 h, and 24 h). A densitometric assay of the p-AKT/total AKT ratio was performed. (**B**) The CB1R (AM251, AM281) and CB2R (JTE-907) antagonists’ effects on the p-AKT/total AKT ratio with AEA (5 µM) at 24 h. The CB1R and CB2R antagonists AM281, AM251, and JTE-907 were employed as pre-treatments before AEA (5 µM). A densitometric assay of the p-AKT/total-AKT ratio was performed. ^®^-actin was used as a loading control. The densitometry analysis of three independent experiments on the p-AKT/total AKT ratio is represented as mean values ± S.E.M. One-way ANOVA was followed by Dunnett’s test; * *p* ≤ 0.05 and ** *p* < 0.01, differently from the control (DMSO). Tukey’s test; *^#^ p* < 0.01, differently from the control (DMSO) and AEA+JTE-907. Graphical material was created using GraphPad Prism v. 9.5. (**C**) Nuclear p-AKT immunofluorescence, DAPI (nucleus staining), and phalloidin (actin stress fiber staining) were performed on an AGS gastric cancer cell line (scale bars, 100 µM). The cells were treated as indicated in the methods and then processed for immunofluorescence. Columns, from left to right, include phalloidin, DAPI, p-AKT, and merge staining. Images were obtained using a 40× objective.

**Figure 2 ijms-26-02033-f002:**
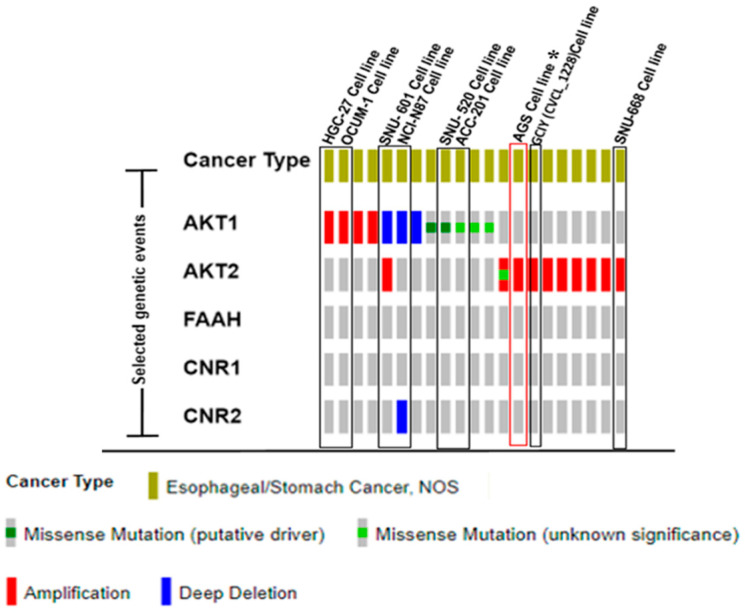
Oncoprint genomic database of esophageal/stomach cancer cell lines. Data query about gene alterations included AKT1, AKT2 (serine/threonine kinase 1/2), endogenous cannabinoid receptors, CB1R and CB2R, and FAAH (hydrolase degradation enzyme) in AEA. Color bars represent specific genomic alterations as amplifications, missense mutations, and deletions. * AGS cell line included AKT1 amplification as specific modification compared to other cell lines analyzed.

**Figure 3 ijms-26-02033-f003:**
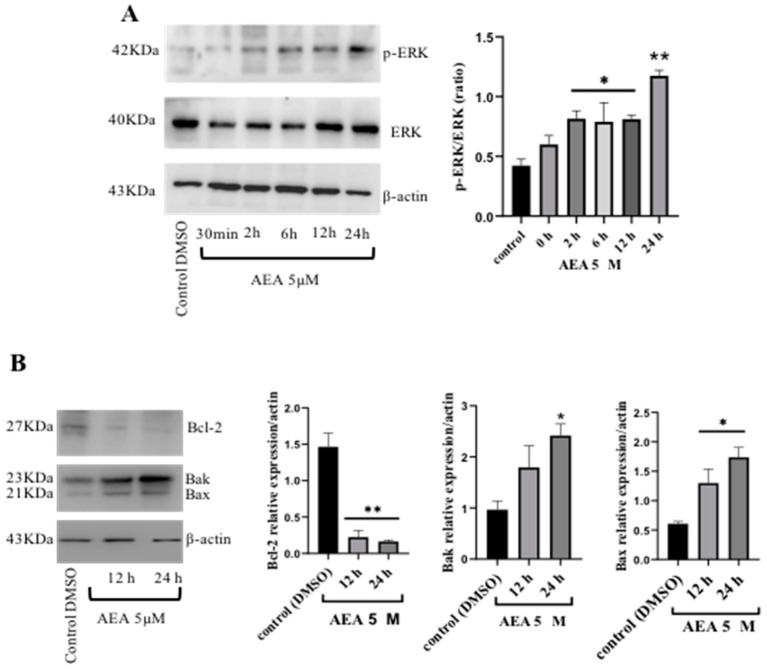
Changes in phosphorylated ERK (p-ERK) in AGS cell lines induced by AEA (5 µM). (**A**) The up-regulation of p-ERK induced by AEA (5 µM) at different moments of time (30 min, 2 h, 6 h, 12 h, and 24 h). A densitometric assay of the p-ERK/total ERK ratio was performed. ^®^-actin was used as a loading control. Densitometry analysis of three independent experiments on the p-ERK/total ERK ratio is represented as mean values ± S.E.M. One-way ANOVA was followed by Dunnett’s test; * *p* ≤ 0.05 and ** *p* < 0.01, differently from the control (DMSO). (**B**) The anti-apoptotic Bcl-2 expression level and pro-apoptotic Bax/Bak were determined by Western blotting in AGS gastric cancer cells treated with AEA (5 µM) for 12 and 24 h. Actin was used as a loading control. Changes in the Bcl-2 (anti-apoptotic), Bak, and Bax (pro-apoptotic) expressions in the AGS cell line induced by AEA (5 µM). Densitometry analysis showing the decreased expression of the anti-apoptotic protein Bcl-2 in AEA-treated AGS cells at 12 and 24 h; meanwhile, an increase is observed in the pro-apoptotic proteins Bax/Bak at the same times. ^®^-actin was used as a loading control. All data are expressed as the mean ± one S.E.M. of three independent experiments. One-way ANOVA was followed by Dunnett’s test; * *p* < 0.05 and ** *p* < 0.01, differently to the control. Graphical material was created using GraphPad Prism v. 9.5.

**Figure 4 ijms-26-02033-f004:**
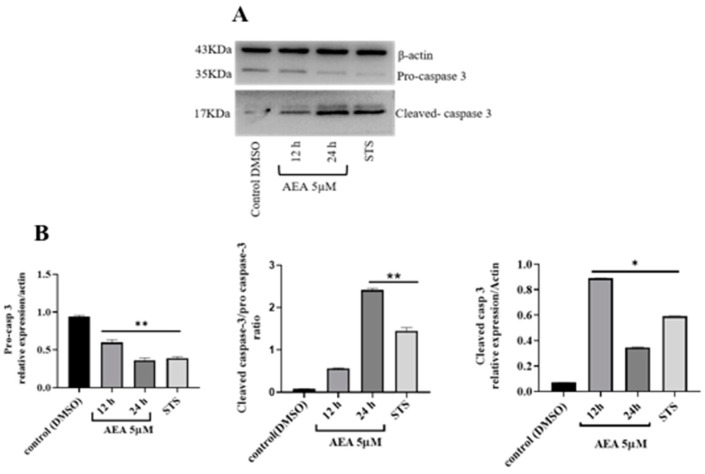
Apoptotic cell death in AGS gastric cancer cell line induced by AEA (5 µM) (**A**) Caspase-3 activity in AGS treated with AEA (5 µM). The AGS cancer cell line was treated with AEA (5 μM) for 12 and 24 h, with DMSO as a control. The protein kinase A inhibitor staurosporine (0.5 µM) was used as a positive control of cell death (2 h). (**B**) Densitometry analysis of cleaved pro-caspase-3, cleaved caspase-3/pro-caspase-3, and cleaved caspase-3 was performed using representative photographs of experimental Western blots. Data are expressed as the mean ± one S.E.M. of three independent experiments. One-way ANOVA was followed by Dunnett’s test; * *p* < 0.05 and ** *p* < 0.01, differently to the control.

## Data Availability

Data contained within the article.

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
