# Peer review of "Pro-Apoptotic Effects of Anandamide in Human Gastric Cancer Cells Are Mediated by AKT and ERK Signaling Pathways"

_ijms, 2025, doi:10.3390/ijms26052033_

Round 1

Reviewer 1 Report

Comments and Suggestions for Authors

This is an extention of the study on the effect of AEA, a member of ECS. The study is interesting, but it is a single cell line study, especially AKT1 mutant gastric cell line.  Please address the following points:

1. The signaling pathway here presented is specific for the cell lines having AKT1 maplification? 

2. Can p-Akt be devided into p-AKt1,2,3? What is the phosphorylated residue in Akt here?

3. Ref 13 looks like on colon cancer cell lines. Are there any other information on AEA effects on gastric cancer cell lines?

4. 5 microM of AEA is used, but any dosage-dependent effet on anti-proliferative activity?

5. Most importantly, do we have the information on physical concentration of AEA around gastric microenvironment?

6. Another important point would be whether the phenomenon the authors observed is specific to the gastric cancer cell lines or any cancer cell lines with or without AKT alterations.

7. The addition of the data using AKT non-altered cancer cell lines would improve the charmness of the manuscript.

Author Response

Comment 1. The signaling pathway here presented is specific for the cell lines having AKT amplification.

Response 1. The AKT and ERK signaling pathways are not specific for cell lines having AKT amplification. Alterations on AKT isoforms (1, 2, and 3) occur in 3 to 5% of cancer cell lines. Besides, other modifications such as mutations and changes in transcriptional AKT genes could be activated simultaneously, which makes oncogenic phenotypes more complex. AKT pathways involve trophic environment effectors directing proliferation and survival in cancer (Sementinio et al. 2024; Kyung et al, 2015). In this study, AKT amplification could be relevant in AEA-treated cells but this needs more studies. A) Kyung, H.Y., & Lauring, J (2015). Recurrent AKT mutations in human cancers: functional consequences and effects on drug sensitivity. Oncotarjet, 7(4),4241. B) Sementino, E. Hassan. D., Bellacosa, A., & Testa, J.R. (2024). AKT and the Hallmarks of Cancer. Cancer Research, 84 (24), 4126-4139.

Comment 2. Can p-AKT be divided into p-AKT 1,2,3? What is the phosphorylated residue in AKT here?

Response 2.  We employ the phosphorylated AKT product; sc-514032 provided by Santa Cruz Biotechnology. Technical details indicate that this product is capable of identifying the three isoforms of phosphorylated AKT at serine residues 473,474 and 472 respectively. In this case, the first dark band in the experimental western blot is AKT1 at Serine 473.

Comment 3. Ref. 13 looks like colon cancer cell lines. Is there any other information on AEA effects on gastric cell lines?

Response 3.  The Ref 11. focuses on the impact of AEA on gastric cell lines. They observed that the AEA effects were stronger when paired with Paclitaxel. Ref. 29 related with WIN-55,212 synthetic cannabinoid on gastric cancer cells, induced proliferation, and cell cycle inhibition (Ref. 11 Miyato H.et al. Pharmacological synergism between cannabinoids and paclitaxel in gastric cancer cell lines. J.Surg Res. 2009;155:40-57), (Ref.29 Xian X. et al. Effect of a synthetic cannabinoid agonist on the proliferation and invasion of gastric cancer cells. J.Cell Biochem; 110:321-32.

Comment 4. 5 microM of AEA is used, but any dosage-dependent effect on anti-proliferative activity?

Response 4. Previous studies were done on this issue. Ref. 20 focuses on the effects of three cannabinoids (AEA, Math-AEA, CP-55,940) employed at different concentrations on AGS cancer cell line evaluating mitochondrial metabolism.  AGS viability decreased by around 60% using 5 microM of the three agonists. (Ref. 20 Ortega, A., Garcia-Hernandez, V.M., Ruíz García, E., Meneses-Garcia, A., Herrera- Gómez, A., Aguilar, Ponce, J.L. Del Angel S.A. (2016). Comparing the effects of endogenous and synthetic cannabinoid receptor agonists on survival of gastric cancer cells. Life Science, 165, 56-62).

Comment 5. Most importantly, do we have information on the physical concentration of AEA around the gastric microenvironment?

Response 5. The physiological concentration of AEA in the gastric microenvironment is around 1-5 nM (Hillard et al, 2018; Cañumil et al, 2023). AEA levels increase in tumor leucocytes. The tumoral microenvironment is represented by immunity cells which express endogenous cannabinoids and receptors, known as the "Immunitary endocannabinoid system", but this issue needs more study (Rahaman et al, 2021; Kienzl et al, 2020) A) Hillard C.J. Circulating endocannabinoids: ?from whence do they come and where are they going? Neuropsychopharmacology 2018. 43:155-72. B) Cañumil, V.A. de la Cruz Borthiry, F.L., Scheffer, F., Herrero, Y., Scotti, L. Boguetti, M.E. A physiological concentration of anandamide promotes the migration of human endometrial fibroblast and the interaction with endothelial cells in vitro. Placenta, 139, 99-111. C) Rahaman, O., & Ganguly, D, 2021. Endocannabinoids in immune regulation and immunopathologies. Immunology, 164,(2)., 242-252. D) Kienzl, M., Kargl J., & Schicho, R. 2020. The immune endocannabinoid system of the tumor microenvironment. International Journal of Molecular Science. 21 (23), 8929.

Comment 6. Another important point would be whether the phenomenon the authors observed is specific to gastric cancer cell lines or any cancer cell lines with or without AKT alterations.

Response 6. Several researchers have observed that the cannabinoid effects are specific in tumor cells independently if they have or no AKT alterations, this is due to cell death molecules activity mediated by alternative signaling pathways. Meanwhile, other researchers have demonstrated that the cannabinoids have protective activity on normal cells (primary cell cultures). Also, in cancer, other mutations different than AKT could act as main effectors of malignancy, such as KRAS, p. Gly12Asp (c. 1357>A); PIK3CA, p.Glu453Lys (c.1357G>A); and TP53, (PubMed 1370612).

Comment 7. The addition of the data using AKT non-añtered cancer cell lines would improve the charmness of the manuscript.

Response 7. Our group has other studies in progress that evaluate the AEA effects of non-tumoral cells without AKT  alterations. We do not lose sight that despite AKT alterations tumor progression could be achieved.

Reviewer 2 Report

Comments and Suggestions for Authors

This study demonstrates that anandamide (AEA) inhibits gastric cancer cell proliferation by down-regulating phosphorylated AKT through CB2R activation and inducing pro-apoptotic pathways via phosphorylated ERK. These effects lead to caspase-3 activation and reduced cell viability, highlighting AEA's potential as a therapeutic agent for gastric cancer. I have several questions.

1.What are the specific roles of phosphorylated AKT and phosphorylated ERK in the anti-proliferative effects of anandamide on AGS cells?

2.What additional molecular pathways, aside from those involving AKT and ERK, might be influenced by anandamide in gastric cancer cells?

3.Could the findings of this study be extended to other types of gastric cancer cell lines with different genetic mutations?

4.What is the therapeutic potential of combining anandamide with other treatments targeting the AKT or ERK pathways?

5.How might the observed effects of anandamide on AGS cells translate to in vivo models of gastric cancer?

6.Are there any known limitations to using endogenous cannabinoids like anandamide as therapeutic agents in gastric cancer treatment? 

7.It is known that various ion channels, such as TRPM7, are involved in the antiproliferative regulation of gastric cancer cells. It is recommended to either conduct experiments related to these ion channels or discuss their relevance in the discussion section.

Author Response

Comment 1. What are the specific roles of phosphorylated AKT and phosphorylated ERK in the anti-proliferative effects of anandamide on AGS cells?

Response 1. When p-AKT signaling is inhibited, it can lead to the activation of pro-apoptotic factors and deactivation of anti-apoptotic proteins such as Bcl-2. This shift in balance favors cell death. The balance between the intensity and duration of pro- and ati-apoptotic signals transmitted by ERK1/2 may determine cell proliferation or apoptosis. Also, p-AKT promotes cell cycle progression, angiogenesis, migration, and differentiation by mTOR activation. mTOR activates S6K1 and 4EBP1 increasing ribosomal proteins (Matsuoka et al, 2014). p-ERK regulates transcription effectors as: ETS, MYC, and ELK direct cell death and metabolism (Pandian et al 2022) A) Matsuoka, T., & Yashiro, M. 2014. The role of PI3K/Akt/mTOR signaling in gastric carcinoma. cancers, 6 (3), 1441-1463. B) Pandian, J & Ganesan, K. 2022. Delineation of gastric tumors with activated ARK/MAPK signaling cascades for the development of targeted therapeutics. Experimental Cell Research, 410 (1), 112956.

Comment 2. What additional molecular pathways, aside from those involving AKT and ERK, might be influenced by anandamide in gastric cancer cells?

Response 2.  The ceramide pathway is another mechanism by which AEA induces cell death. Is well known that AKT inhibition increases ceramide levels driving toward cell death molecules co-activation.

Comment 3. Could the findings of this study be extended to other types of gastric cancer cell lines with different genetic mutations?

Response 3. Yes, because different gastric cancer cell lines share AKT alterations with crosstalk scaffold proteins activated by Tyrosine Kinase receptors of the immunoglobulin family (EGFR, VEGFR, IGFR, etc).

Comment 4. What is the therapeutic potential of combining anandamide with other treatments targeting the AKT or ERK pathways?

Response 4.  There are clinical reports that demonstrate the antiproliferative effects of AEA in combination with 5-FU, paclitaxel, and other therapeutic agents in several cancer cell models ("in vitro" and nude animals).

Comment 5. How might the observed effects of anandamide on AGS cells translate to in vivo models of gastric cancer treatment?

Response 5. Animal xenografts (rats or mice) transplants or grafts of gastric tumor cells. AEA in this model induced a diminishment of tumor volume. These antiproliferative effects are improved using classical treatments for gastric cancer.

Comment 6.  Are there any known limitations to using endogenous cannabinoids like anandamide as therapeutic agents in gastric cancer treatment?

Response 6. FAAH (Amide hydrolase enzyme) degradation enzyme of endogenous anandamide might be a key target to know if there are any limitations, but this field is starting to be explore.

Comment 7. It is known that various ion channels, such as TPRM7, are involved in the antiproliferative regulation of gastric cancer cells. it is recommended to either conduct experiments related to these ion channels or discuss their relevance in discussion section.

Response 7. Experiments related to ionotrophic channels could be explored in our following studies. Ionotrophic signals and their effects on gastric cancer are limited. A)Li Li, Cheng Chen , Chngyao Chiang, Tian Xiao, Yangchao Chen, Yongxiang Zhao, Duo Zheng, the impact of TRPV1 on cancer pathogenesis and Therapy: A systematic review, 2021 May 11;17(8): 2034-2049.

Round 2

Reviewer 1 Report

Comments and Suggestions for Authors

The revisions are acceptable.

Reviewer 2 Report

Comments and Suggestions for Authors

It is well revised.